# The Role of Bioactive Phenolic Compounds on the Impact of Beer on Health

**DOI:** 10.3390/molecules26020486

**Published:** 2021-01-18

**Authors:** Roberto Ambra, Gianni Pastore, Sabrina Lucchetti

**Affiliations:** Council for Agricultural Research and Economics, Research Centre for Food and Nutrition, 00178 Rome, Italy; giovanni.pastore@crea.gov.it (G.P.); sabrina.lucchetti@crea.gov.it (S.L.)

**Keywords:** beer, phenols, alcohol, health

## Abstract

This review reports recent knowledge on the role of ingredients (barley, hop and yeasts), including genetic factors, on the final yield of phenolic compounds in beer, and how these molecules generally affect resulting beer attributes, focusing mainly on new attempts at the enrichment of beer phenols, with fruits or cereals other than barley. An entire section is dedicated to health-related effects, analyzing the degree up to which studies, investigating phenols-related health effects of beer, have appropriately considered the contribution of alcohol (pure or spirits) intake. For such purpose, we searched Scopus.com for any kind of experimental model (in vitro, animal, human observational or intervention) using beer and considering phenols. Overall, data reported so far support the existence of the somehow additive or synergistic effects of phenols and ethanol present in beer. However, findings are inconclusive and thus deserve further animal and human studies.

## 1. Introduction

Beer is a natural drink and historical evidences indicate a common use since ancient times also for medical and religious purposes [1]. Antique recipes prove widespread production back to 5000 years ago [2]. Beer is actually the most consumed alcoholic beverage in the EU and annual per capita consumption (L/year) has sharply increased in the Czech Republic (141 L), US (50–80 L) and France (33 L) [3]. Such a level of consumption has led some research to focus on the nutritional appropriateness of beer, merely considering health aspects like, for example, the intake of minerals [4] or the ability to prevent dysbiosis [5], properties also present in other beverages. Unfortunately, like wine, beer naturally contains ethanol, a well-known toxic and carcinogenic molecule [6].

Nonetheless, characteristic of beer is the high content in phenolic compounds, which are the focus of this review. The consumption of polyphenol-rich foods, like beer, is a well-accepted factor involved in the prevention of oxidative stress-associated diseases [7]. Traditionally, beer is obtained from as little as four basic ingredients: barley, hop, yeast and water. The first two ingredients naturally contain phenolics, however during beer production, these molecules undergo chemical modifications and new molecules are formed, influencing both the yield and final characteristics of a beer. Aroma, flavors, taste, astringency, body and fullness are the result of the metabolic activity of microbes on raw materials, and scientific evidences suggesting that they are all influenced by phenol content are summarized here. Moreover, this review focusses more deeply on most recent advances on the role of phenolic compounds on affecting human health status, considering how seriously researchers have tackled the effects of alcohol.

## 2. Main and Minor Beer Phenols

The polyphenolic composition of beers is considered as one of the quality indicators of beer processing and marketing [8]. In fact, the type and quantity of phenols influence taste, aroma and color, but also colloidal and foam stability, shortening beer’s shelf-life taste (see Section 4, “Phenols and beer attributes”). Several different groups of phenolic compounds have been reported in beer, the main ones being phenolic acids and tannins, and flavones and flavonols [9]. Because of its high concentration, also thanks to high producing yeasts (see Section 5, “The role of barley, yeast and hop genetics on beer phenols”), the simple phenolic alcohol tyrosol is one of the main phenols looked at in beer, present also in alcohol-free beers [10]. Concentration is so high in certain beers, reaching that of red wine [11], that authors have hypothesized that tyrosol could represent an indirect source, through biotransformation, of the more biologically active hydroxytyrosol [12] (see Section 6, “Phenols-related health effects of beer consumption”). In alcoholic beers, both phenols possibly protect yeast from the stress generated by high levels of ethanol, a phenomenon that has been demonstrated for wine’s resveratrol [13], indicating that phenols not only undergo changes during brewing, but they also direct it. Accordingly, non-alcoholic beers normally have lower phenolic content [14], supporting the existence of a correlation between phenols and alcohol concentrations. Among minor phenols, those derived from barley, for example alkylresorcinols, are a group of phenolic lipids for which in vitro antioxidant and antigenotoxic [15] and in vivo diet-induced obesity-suppressing [16] activities have been reported. Even if contribution to alkylresorcinols dietary intake appears not significant, higher amounts were reported in stout beer [11]. Other quantitatively minor phenols derived from hop, for example, xanthohumol and other prenylated flavonoids, contribute significantly to beer flavor and aromas and have antibacterial, anti-inflammatory and antioxidant properties, and phytoestrogen activity [17,18]. Prenylflavonoids are of particular interest for beer as, on the one hand, no other food sources other than hop are known and, on the other hand, they are present regardless of the fermentation method, ale or lager, even if higher concentrations were found in stout and India Pale Ale styles [11].

Despite the fact that prenylated flavonoids can last for 10 years in beer stored at room temperature [19], monophenols and flavonoids show a temperature- and time-dependent decay in beer [20,21]. This phenomenon was initially studied using radioactive isotopes that revealed that almost 65% of molecules belonging to the tannin fraction go through oxidation [22]. Later, other evidences supported the role of oxidation in the time-dependent decay of phenols in beer, also demonstrating the role of the intrinsic haze-forming ability of some phenols [23]. Meanwhile, acetaldehyde was also involved in haze formation, because of its ability to polymerize polyphenols and compromise beer’s flavanols level [24]. A resolutive approach to this problem could come from the implementation of dry-conservation. It was recently reported that production of microparticles from beer through high-temperature (up to 180 °C) spray-drying, used for the development of functional food with a specific heath objective, yielded a well-accepted beverage, in terms of appearance, taste and color, that kept, up to the entire period of dry-conservation (180 days), the initial amount of total phenols (measured using the Folin–Ciocalteu method) [25]. Even if no qualitative indication of phenols was reported, the study supports the validity of spray-drying in the production of non-alcoholic, high-phenols, beer-flavored beverages (see Section 9, “Phenols in non-alcoholic and isotonic beers”).

## 3. Phenols’ Fate during Malting and Brewing

As mentioned in the introduction, beer content in phenols depends on the type of barley and hops used for production. Even if hops contain a huge amount of phenols (up to 4% of dry matter) compared to barley (up to 0.1%), on average, four fifths of beer’s phenols come from malt or other mashed cereals, because of their significantly higher starting amount [26]. Phenols undergo both quantitative and qualitative changes during seed germination and brewing processes [27] (Figure 1). The germination of barley seed, i.e., malting, has been studied deeply and is preceded by seed hydration (steeping), during which phenolic content decreases due to leaching, and followed by seed-drying (kilning), during which the improved crumbliness of the grain enhances the enzymatic release of bound phenolic acids. Kilning can be performed at different temperatures, for example in special malts brewing in order to bring desirable flavors and colors [28]. At temperatures lower than 80 °C, kilning normally induces an increase in the amount of water-soluble total phenolic compounds [29], thanks to a Maillard-enzymatic release of phenols in the matrix [30] and to increased friability and extraction from the grain [31]. According to Leitao and colleagues, total phenolic content of barley (whose antioxidant contribution is mostly for ferulic and sinapic acids) increases four-fold during the transition to malt. Even if final yields depend on the malting procedures, the amount of phenolic compounds present in malt is inversely correlated with the degree of steeping and positively influenced by the germination temperature [32]. More recently, Koren and coworkers reported a 3- to 5-fold increase in the amount of total polyphenols during malting in six barley varieties, independently from the initial amounts [33].

The amount of polyphenols reached in malt then significantly falls during brewing steps, depending on the protocol adopted, with a higher decrease for malt milled in wet conditions [34]. Enzymatic and non-enzymatic solubilization of phenols take place during the first step of mashing (hot hydration), and both are influenced by temperature and time, as well as the separation of wort, during which extraction of phenolic-rich spelt material occurs [9]. A successive increase of total phenolic compounds occurs in the wort separation (lautering) due to the extraction from spelt materials. Brewing is fundamentally ascribable to the metabolic activity of a fermentable carbohydrate source in the absence of oxygen, yielding alcohol and carbon dioxide. Fermentation is normally performed at fixed temperature but can be pushed at higher or lower temperatures. Hops, which were formerly included in the brewing process mainly for their preserving properties, are then added and wort boiling is started. Hops addition actually has several advantages, improving not only the bitter taste and astringency but giving protection to beer brewing yeasts, thanks to its antibacterial activity, against Gram-positive bacteria, and lowering pH to 4–4.2 [35]. During boiling, hop polyphenols are released and polymerization reactions with proteins occur, yielding precipitated complexes, responsible for the formation of chill haze, that are then lost in the successive whirpool process and during the final filtration and stabilization. Final processes are critical for polyphenols and include fermentation, warm rest, chill-lagering filtration and clarification [36]. During brewing, around 60% of the malt phenolic content is lost. Decay affects all phenolic compounds, excepting *p*-hydroxybenzoic acid and sinapic acid, whose concentration increases by even four-fold [31]. However, different brewing processes can deeply influence total phenolic compounds, for example bock beers are normally three times richer than dealcoholized beer, with intermediate and decreasing quantities for abbey, ale, wheat, pilsner and lager beers [36]. Recent data also indicate that beer’s content in phenols is associated with the production scale. In fact, the lesser characterized craft beers (unpasteurized and unfiltered) [37], whose production scale is limited by law in several countries (200,000 hL/year in Italy), exhibit higher total phenolic compounds’ values compared to large-scale beers [38], mainly thanks to the lack of filtration. Finally, the phenolic content of beer is affected negatively by higher temperature pasteurization treatments [39]. 

## 4. Phenols and Beer Attributes

The ability of phenols to influence beer taste has been well known since the early 1960s, when the so-called “sunlight flavor” was ascribed mainly to humulone and lupulone addition after beer fermentation [40]. Phenols’ ability to interfere with aroma, instead, was noticed around forty years ago, thanks to a *S. cerevisiae* “killer strain” producing a clove-like aroma [41]. Later, a study clarified that presence of the main phenolic flavors relies on yeasts capability to decarboxylate or reduce phenolic acids: 4-vinylguaiacol and 4-vinylphenol from *S. cerevisiae* and 4-ethylguaiacol and 4-ethylphenol from *Brettanomyces* sp. [42]. More recent data indicate that the ability of phenols to selectively characterize beer’s flavors relies on their chemical transformations. For example, thermal decarboxylation of ferulic acid to 4-vinyl guaiacol, occurring during wort boiling and during fermentation, induces a three-orders-of-magnitude increase in its flavor threshold [43]. Unfortunately, some metabolic reactions have side effects, like that involving cinnamic acid and yielding the toxicologically relevant styrene [44]. Moreover, higher concentrations of monophenol can turn spicy or vanilla-like sweet flavor notes to unpleasant medicinal-like flavors [45]. A recent deep analysis of the association between metabolites and sensory characteristics using two-way orthogonal partial least squares indicates that isoferulic acid affects beer’s fruity sensory attributes [46], suggesting the possibility to predict to some extent the formation of specific flavors.

With respect to aroma, phenols’ protecting properties were found almost 25 years ago: phenols were found to prevent the formation of off-flavors, before and during malting, and the phenomenon was ascribed to their antioxidant activity in barley and malt [47]. More recently, some specific monophenols that confer the typical aroma of some popular beers were identified [48] and recently reviewed [49]. Worthy of interest are Czech beers whose distribution of individual phenolic compounds, that has been brought back to the origin of raw materials and the technology used for processing, is so unique that they have been proposed for authenticity analysis [50,51]. With respect to color, after high-affinity selective removing of tannins, Dadic and Van Gheluwe observed a severe discoloration of beer, demonstrating for the first time the correlation between phenols and beer color [52]. The involvement of monoflavanols’ oxidation on beer color was further demonstrated by the recovery of oxidized molecules in polyethylene terephthalate bottle-stored beer [20]. More recently, several works have clearly demonstrated the relationship between phenols and beer color, both in small- and large-scale brewed beers [38]. 

Barley seeds’ phenolic acids, flavonoids and proanthocyanidins influence quality indexes like viscosity, diastatic power and nitrogen content [53], and have an impact on beer turbidity [54], taste, bitterness and aroma [55]. With regard to hop, which was antiquely added in beer especially for its pleasant aroma and bitterness, brewing trials indicate that hop phenols can selectively reduce flavor deterioration during storage [56], specifically the sunstruck off-flavor that is formed in beer upon light exposure [57]. More recent data indicate a temporal effect. In fact, later addition of hop, just before the end of wort boiling, significantly increases phenolic content [58]. Astringency, bitterness and fullness, which are affected by the boiling time [39], have been linked to different hop phenols fractions [59,60]. 

## 5. The Role of Barley, Yeast and Hop Genetics on Beer Phenols

The yield in phenols of a beer necessarily depends on the genetic background of its raw ingredients, and differences were reported in barley grain [61], hop [62] and yeast [63]. Unfortunately, domestication of barley and hop has reduced phenols’ diversity. Nevertheless, total polyphenol content could be linked to specific quantitative trait loci in barley [64] and some specific combinations of phenols in barley can still be attributed to different genotypes. For example, the ratio between barley’s main phenolic acids, ferulic acid and *p*-coumaric acid, is genetically determined and combinations can also influence key agronomic traits, such as hull adherence and grain color [65], through functionally related genes [53]. Studies combining genetics and environment on wild barley cultivars, that show a wider genetic diversity in agronomic traits and abiotic stress tolerance, identified some genes involved in phenol accumulation in barley seeds. Such studies are of special relevance as they can give a picture of the loss of genetic variation due to domestication and provide information for the set-up of breeding applications for phenols-related beer improvement. For example, a network analysis of gene expression and secondary metabolites, induced by the well-known stressor drought [66] in developing grains from several different Tibetan wild barley cultivars, recently allowed the identification of genes whose manipulation is believed to help the development of cultivars with specific contents of phenolic compounds [67]. Less data is available for a role of the genetic background on hop phenols. For example, a significant cultivar-dependent role has been recently reported for 2-phenylethyl glucoside [68], but the relevance on final quantities recoverable in beer is still lacking.

The ability of yeasts to adapt to different chemical (sugar, nitrogen) and physical (temperature, pH, oxygen, sulfur dioxide) properties resides in the great genetic diversity that has been exploited by the beer industry, i.e., for the development of strains with distinct flavor profiles. The production of different metabolites, like volatile phenols, is the direct consequence of human influence through wine and beer production. A first evidence testifying the role of the genetic background of yeasts in beer phenols came from the observation, at the beginning of the twentieth century, of volatile “ethereal substances” in English stock ales, during fermentation by Brettanomyces [69]. *Brettanomyces bruxellensis*, the first microorganism to be patented for beer production, was also involved in the spoilage of draught beer [70] and in the clove off-flavor (the ethylphenol 4-vinylguaiacol) [71] but, after being reported together with *Lactobacillus vini* as a contaminant in several ethanol-producing plants [72], was finally isolated from a number of fermented beverages and food, from cider to olives [73]. Spoilage depends on a still not fully identified gene pathway that involves two phenylacrylic acid decarboxylase (PAD) enzymes [74]. Ethylphenols production has been related to strain-dependent PAD amino acid sequence variability [75]. Thanks to their ability to convert ferulic acid to 4-vinylguaiacol, yeasts are believed to have a stronger impact on phenols than thermal processing steps [76]. Yeasts also have a fundamental impact in barrel beer ageing. Barrel-aged beers are sensorially enriched beers obtained by storage of already fermented beers in wood casks or by fermentation of beer’s wort directly in wood barrels. Such processes mainly occur because of the spontaneous growth of microbes present in breweries’ atmosphere and in barrels [77]. During this fermentative incubation, a bi-directional exchange of different molecules occurs from wood and beer: some beer’s molecules are retained by the wood while others are released from wood to the beverage. *Dekkera bruxellensis*, another spoilage-related microbe in wine, is considered the main contributor to the aroma of aged beers, through its ability to convert hydroxycinnamic acids to volatile phenols, and has several advantages, from high ethanol yield to low pH tolerance [78]. Its spontaneous growth is accompanied by some enzymatic activities that transform wort composition and yield the final chemical and sensory profiles of aged beer.

Aiming at finding optimal conditions for accelerating wort transformations, research is focused at finding optimal chemical conditions to produce beers with specific and preferred bacterial metabolites, normally avoiding those from non-Saccharomyces species, in multi-starter cultures. For such purpose, Coelho and coworkers recently found that low glucose or high ethanol conditions favor the yield of *D. bruxellensis*-related metabolites over *S. cerevisiae* ones [79]. Ethanol-resistance and increased dominance towards other *S. cerevisiae* strains were also reported on mixed starter fermentations for the high polyphenols-producing *S. cerevisiae* var. *boulardii* strain [80]. A recent deep genomes/phenomes analysis involving 157 industrial *S. cerevisiae* strains [81] reported that production of 4-vinylguaiacol relies on specific genetic variants able to ferment maltotriose [81]. More recently, next-generation sequencing allowed the identification of a Brettanomyces strain void of phenolic off-flavors, limiting economic losses during production [82], a problem that was bypassed in *S. cerevisiae* by the selection of strains with inactivated alleles and/or functional copies [83]. Worth mentioning is a recent work that, seeking to explain different adaptive abilities, profiled microsatellite markers and ploidy-states of 1488 isolates coming from niches dispersed all over the world [84].

## 6. Phenols-Related Health Effects of Beer Consumption

While the serious damages of high alcohol intake are known, the effects of moderate consumption of alcoholic beverages are still a source of heated debate. Moderate beer consumption is believed to be associated with protective cardiovascular function and reduction in the development of neurodegenerative disease. Moreover, there is no evidence that moderate beer consumption can stimulate cancer. Nevertheless, alcohol consumption can become a problem for people at high risk of developing alcohol-related cancer or for those affected by cardiomyopathy, cardiac arrhythmia, depression, liver and pancreatic diseases, and is not recommended for children, adolescents, pregnant women and frail people at risk of alcoholism [85]. Anyway, beer, like wine, contains the already mentioned substances with indubitable protective capacities, not merely anti-inflammatory and antioxidant, as demonstrated by huge in vitro work on single substances [86]. However, the ambitious objective in studying the effects of beer consumption on human health is to analyze it in toto and, in order to understand the single contribution of phenols and alcohol, parallel experiments with similar doses of an equivalent non-alcoholic beer and of alcohol alone are essential. For example, Karatzi and coworkers [87] reported that both non-alcoholic and alcoholic beers improved some arterial biomarkers (reduced aortic stiffness and increased pulse pressure amplification), but the effects were also similar in a parallel vodka intervention, containing the same amount of ethanol as the alcoholic beer. However, as some other effects (wave reflections reduction) were higher in the alcoholic beer intervention compared to alcohol alone (vodka), and the endothelial function was significantly improved only after beer consumption, the authors concluded that the non-alcoholic and the alcoholic fractions of beer could have additive or synergistic effects [87]. 

We thus thought to analyze the fraction of similar publications that considered, in the search of the health effects of beer containing phenols, also the effects of the presence of alcohol. For such purpose, we used in Scopus.com the search string TITLE-ABS-KEY (beer AND (phenol OR polyphenol OR flavonoid) AND (observational OR administration OR consumption OR drinking OR prospective OR intervention OR crossover OR trial)) AND (LIMIT-TO (DOCTYPE, “ar”)). The search was performed on October 2020 and returned 161 documents, including 31 reviews (even if they were already excluded by the string search), 7 not pertinent articles, 9 studies merely evaluating phenols’ population intakes, 51 chemical-only reports (papers reporting chemical analyses of phenols of commercial or improved beers) and 22 reports using only single phenols in in vitro or in vivo models. For the remaining 41 (minus one not available even by the authors themselves [88]), experimental models, parameters tested and main findings are summarized in the next section and sorted chronologically by the most recent, in Tables 1–3, about in vitro and animal models, human intervention and human observational, highlighting the use of alcohol alone (spirits, eventually vodka or gin), as well as non-alcoholic beer.

### 6.1. In Vitro and Animal Experiments

As demonstrated by in vitro cancer cell models (Table 1), several cancer types are sensitive to the antiproliferative action of some beer components, including ethanol. For example, epithelial cells’ viability was reduced in a similar way by beer or an equivalent amount of ethanol [89]. Unfortunately, the authors did not test an alcohol-free beer. Using single molecules or a matrix containing all beer components, Machado and coworkers showed that phenols’ activities are synergic [90]. Unfortunately, in this case, ethanol was not tested. Similarly, a total extract obtained from dark beer conferred higher protection to rat C6 glioma and human SH-SY5Y neuroblastoma cells against an oxidant stressor challenge (hydrogen peroxide) compared total extracts obtained from non-alcoholic and lager beers [91]. Again, neither the phenolic compounds of beers nor an alcoholic reconstituted extract were tested.

Wistar rats were used in several experiments with beer. One publication reported that both administration of alcoholic (4%) or lyophilized beer for 4 weeks had low, but statistically significant, beneficial effects on plasma lipidemic and antioxidant markers (total cholesterol, low-density lipoprotein (LDL) cholesterol, triglycerides and lipid peroxides), however alcohol alone was not tested and the authors themselves concluded that minimal effects observed could rely on relatively low alcoholic content of beer [100]. Next, using only a polyphenol-free beer, the same group concluded that lipid effects had to be ascribed to beer proteins, as long as effects were absent in rats fed with polyphenol-free wine [99]. In rats with skin incision-induced wound healing, feeding for 4 weeks with alcoholic beer prevented alcohol-induced markers of inflammation, oxidative stress and angiogenesis [97]. Notably, when beer was enriched with 10 mg of xanthohumol, effects were even more ameliorated. Similar results were obtained using animals with streptozotocin-induced diabetes [96]. On the same streptozotocin-induced diabetes model, hepatic glucolipid metabolism, lipogenic enzymes and glucose transporter 2 levels were tested after 5 weeks of administration of xanthohumol-enriched alcoholic beer for 5 weeks [95]. Interestingly, beer prevented all the streptozotocin-induced liver catabolic state alterations tested (fibrosis, apoptosis, glycogen depletion, GLUT2 upregulation, lipogenesis reduction) and the effect was not observed in rats fed with normal beer. The authors also tested the effect of ethanol alone but, in none of these last three works were an alcohol-free beer, nor xanthohumol alone, tested, thus it is impossible to distinguish neither the effect of beer components nor of the polyphenol itself. Furthermore, in vitro and in vivo work on xanthohumol metabolites (isoxanthohumol and 8-prenylnaringenin) previously indicated opposite effects on angiogenesis and inflammation processes (pro-angiogenetic for 8-prenylnaringenin and anti-angiogenic and anti-inflammatory for the other two) [101]. Nevertheless, a xanthohumol-fortified alcoholic beer was used again to demonstrate attenuated pharmacologically induced pulmonary vascular remodeling and improved cardiac function [93]. Also, in this case, even if effects were absent in rats fed only with ethanol, no rats were tested with an alcohol-free beer. It is noteworthy that the authors could identify the involvement of extracellular signal-regulated kinase1/2, phosphatidylinositol 3-kinase/protein kinase B and VEGF receptor 2 in the protective properties of beer towards pulmonary arterial hypertension [93]. In a prepubertal rat model, beer with 10% alcohol significantly decreased, after 4 weeks, the levels of sex hormones, compared to ethanol- or water-fed rats [92]. Again, even if authors concluded that beer inhibited the ethanol-induced increase of cleaved caspase-3 in Leydig cells, a non-alcoholic beer was not tested. In addition to the works recovered using the Scopus.com search string and mentioned in Table 1, worthy of mention are experiments showing that alcoholic-free beer can decrease the aminooxyacetic acid-induced GABA accumulation in hypertensive animals [98], and prevent brain inflammation and neurodegenerative effects induced by aluminum nitrate [94]. However, while as expected hops administration alone had a beer-overlapping positive effects to some extent, so did silicon administration, reinforcing the need for an appropriate set-up of experimental models.

### 6.2. Role of Alcohol on Phenols’ Metabolism and Beer Antioxidant and Anti-Inflammatory Properties, and on Cardiovascular-Related Effects 

Phenolic acids’ absorption, previously reported both in low-alcohol [102] and alcoholic beer [103], is impaired by ethanol removal from beer [104]. The opposite effect of alcohol has been reported for tyrosol metabolization to hydroxytyrosol following beer consumption, as mentioned above. In particular, the administration of a single dose of 250 mL of blonde beer was associated to higher urinary recovery of tyrosol, whilst an identical dose of alcohol-free beer yielded higher urinary recovery of hydroxytyrosol [12]. However, as alcohol consumption proportionally increases hydroxytyrosol excretion through dopamine metabolism [105], hydroxytyrosol bioavailability is hardly attributable only to beer phenols.

Among first beer intervention studies (Table 2), there is an almost-perfectly set-up randomized acute administration of either 4.5% alcoholic beer (*n* = 14), or dealcoholized beer or 4.5% water solution of ethanol (*n* = 7), for the evaluation of the contribution of beer’s alcohol [104]. Results demonstrated that a significant increase in plasma antioxidant capacity (TRAP) could be obtained only following alcoholic beer administration. Unfortunately, no crossover intervention was performed, and the effects were studied only in a temporally limited manner. In another similar, but a crossover, acute intervention of beer or wine (or vodka for the evaluation of the contribution of alcohol) inhibition of oxidative stress induced (by 100% normobaric O_2_ breathing) was tested [106]. Analysis of stiffness 3 h after administration showed that only wine prevented oxygen-induced oxidative stress, possibly because of the higher content of polyphenols compared to beer i.e., 2.6 g/L vs. 0.4 g/L gallic acid equivalents (GAE) [106]. No one can say if such a low phenols amount in an equivalent alcohol-free beer could have produced the effects observed with wine. Daily supplementation of breastfeeding mothers (*n* = 30) with 660 mL of non-alcoholic beer was associated with an improvement of mothers’ plasma and breastmilk antioxidant capacities, assessed 30 days postpartum, compared to control non-supplemented mothers [107]. For obvious reasons, an alcoholic beer was not tested. Administration of alcohol-free beer (500 mL) for 45 days to postmenopausal women (*n* = 29) was associated with a reduction of several indicators of early protein oxidation, especially reducing cholesterol levels in subjects with higher than 240 mg/dL [108], supporting the usefulness of long-term alcohol-free beer consumption in fighting low-grade chronic inflammation and preventing metabolic disorders. As an alcoholic beer was not tested, one might speculate that alcohol can abolish the beneficial effect. However, previous work that used a crossover intervention trial (healthy drinkers, *n* = 27) to switch consumption of beers with similar phenolic content (310–330 mg/L) for 4 weeks, from low (0.9%) to high (4.9%) alcohol and vice versa, indicates that while the switch to low-alcohol did not change in vitro LDL oxidizability, the opposite switch did [109]. On the other hand, only non-alcoholic beer daily consumption for one week (17 healthy females, 330 mL) was associated to an increase in the urinary antioxidant capacity, as measured by Trolox equivalents [110], contradicting the results of the study reported at the beginning of this paragraph.

One observational study (1604 subjects of the IMMIDIET (Dietary Habit Profile in European Communities with Different Risk of Myocardial Infarction: the Impact of Migration as a Model of Gene-Environment Interaction) study, 26–65 years, see Table 3) supports a somewhat interfering property of alcohol on non-alcoholic components of beer. In fact, adjustment of beer intake for alcohol content broke the association between beer consumption and higher plasma and red blood cell omega 3 fatty acids [119]. In the overweight or class 1 obese healthy subjects, the daily consumption of alcoholic beer (but not of alcoholic-free beer with similar amount of total phenols) for four weeks raised HDL levels in subjects with low LDL-lipid profile and facilitated cholesterol efflux from macrophages, without affecting body mass index (BMI), liver and kidney functions, potentially reducing the risk of vessels occlusion by cholesterol deposition [120]. As the consumption of alcohol alone was not tested, it is not possible to exclude that the effects could be at least partially ascribable to alcohol. Similarly, in a crossover study of 28 daily healthy nonsmoking normotensive men consuming alcoholic beer (1125 mL; 41 g alcohol) for 4 weeks, an increase of the awake systolic blood pressure and the asleep heart rate was reported, however the effects were identical in men consuming red wine containing the same amount of alcohol [118], and an alcohol-free beer was not tested. Similarly, analysis of stiffness, 3 h after administration of alcoholic beer or vodka, showed that both protected against oxygen-induced increase in arterial stiffness, making the authors conclude that the observation was probably due to a central vasodilatatory effect of alcohol itself [106]. Again, Gorinstein and coworkers found that alcoholic beer consumption (330 mL daily, containing 510 mg of polyphenols and 20 g of alcohol for 30 days) ameliorated markers of coronary atherosclerosis of hypercholesterolemic in non-drinker males (*n* = 42, 43–71 years) during recovery from coronary bypass surgery [117]. Unfortunately, the control group of the randomized single-blind trial had only water “with minerals of beer”, making it impossible to ascribe effects to either phenols or to alcohol. In a double-blind intervention of healthy male runners (*n* = 277), daily consumption of non-alcoholic beer, for 3 weeks before and 2 weeks after a marathon, reduced interleukin-6 immediately after the race, total blood leukocyte counts immediately and 24 h after the race and post-marathon incidence of upper respiratory tract illness [116]. However, like for breastfeeding mothers mentioned above, alcoholic beer was not tested, we guess for similar obvious reasons. Also, other observational studies (Table 3) suffer from this limitation. For example, a significant inverse association between beer consumption (and not for coffee, nuts, tea, olive oil and red or white wine) and hypertension was found by means of food frequency questionnaires submitted to 2044 adults [121], however neither the consumption of alcohol-free beer nor the contribution of pure alcoholic beverages were evaluated.

An open, randomized, crossover, finely set-up controlled intervention trial of 33 high-cardiovascular risk males drinking daily, for 4 weeks, a non-alcoholic beer (containing 1243 mg of total polyphenols) or an alcoholic beer (containing 1209 mg of total polyphenol and 30 g of ethanol), was repeatedly used (apparently with the same composition of subjects) by a group of Spanish researchers during the last 6 years, to investigate the possible synergistic effects of beer polyphenols and alcohol, using as control an administration of gin (containing 30 g of ethanol). Firstly, in an attempt to use urinary isoxanthohumol as a marker of beer consumption, a similar amount of the metabolite was recovered following non-alcoholic or alcoholic beer consumption, and no excretion was found following gin administration [115]. Notably, group differences in a female sub-population were found, but only an alcoholic beer was tested. Next, they looked for circulating endothelial progenitor cells (EPC) and reported that non-alcoholic beer consumption increased the number of circulating EPCs by 5 units, while in the alcoholic beer group, the increase was 8-fold. However, even if observations were not statistically significant, alcohol alone (gin) induced a 5-fold decrease in the number of circulating EPCs [114], suggesting the existence of some influencing, maybe genetic, factors. Then, they reported that only non-alcoholic beer consumption reduced leukocyte adhesion molecules and inflammatory biomarkers (decreased homocysteine and increased serum folic acid) [113], suggesting a possible antagonistic effect between alcohol and the non-alcoholic fraction of beer. Importantly, the alcoholic beer improved other plasma lipid and inflammation markers (high-density lipoprotein cholesterol, apolipoproteins A1 and A2, and adiponectin) and decreased fibrinogen and interleukin 5, but the effects were ascribed to alcohol as identical effects were observed following administration of a gin dose containing the same amount of alcohol (30 g). Finally, the group of Spanish researchers applied liquid chromatography-coupled Linear Trap Quadropole-Orbitrap mass spectrometry to discover the urinary metabolites produced in the intervention study. Increased urine excretion of hop α-acids and fermentation products were found following beer consumption with respect to the gin administration, but differences were slight and not completely reported [112].

### 6.3. Role of Alcohol on Phenols-Related Effects of Beer on Cancer

A case-control association study (over 14 years) of child acute lymphoblastic leukemia (*n* = 491 + 491) found an inverse relation with maternal moderate consumption (self-reported) of beer (and wine, but not spirits), making authors suggest a protective effect of flavonoids [130]. However, a positive relation was reported also for fathers, which is difficult to explain and minimizes the observation’s reliability. In a similar matched case-control study of drinking/smoking habits (over 10 years) of leukoplakia patients (*n* = 187 + 187; 40–65 years), while a role of regular wine consumption was associated with a decreased probability of disease occurrence (compared to that of spirit drinking that was associated to increased risk), no significant effect for moderate beer drinking was found [126]. The authors concluded that weaker effects of beer were probably due to the different composition in substances synergistically or antagonistically, i.e., polyphenols, interacting with ethanol [126]. Nonetheless, using a 20-country wide one-year (2002) evaluation of alcoholic beverages consumption and total deaths for oral cancer, the same authors estimated a lower risk for beer (and wine) consumers compared to heavy alcohol consumption from spirits [128]. Similarly, the consumption of beer (nor liquor) could not be associated with prostate cancer risk, in a population case-control study taking into account the self-reported alcohol consumption (*n* = 753 + 703; 40–64 years), even if the same authors reported a reduced relative risk associated with increasing level of red wine consumption [129]. More recently, lack of association with prostate cancer was reported for beer consumption (but also for wine and liquor) in a bigger prospective study (*n* = 84,170; 45–69 years) [124]. 

### 6.4. Role of Alcohol on Phenols-Related Effects of Beer on the Microbiota

According to a relationship between microbiota, host genes and diet [131], recent work investigated the possibility that alcohol-free beer, acting at the level of gut microbiota, could prevent the metabolic syndrome (MS). In fact, occurrence of MS can be promoted by gut microbiota dysbiosis, through low-grade inflammation and alteration of lipid metabolism. Gut microbiota dysbiosis can in turn be induced by alteration of the relative abundance of bacterial families [132]. Thus, a daily administration for one month of 355 mL of non-alcoholic beer was found associated to a decrease in fasting blood serum glucose and an increase in functional *β*-cells only, and the effect was not observed with an alcoholic beer containing a similar amount of phenolic compounds [111]. Moreover, the authors observed an enrichment of the microbiota diversity, also with the alcoholic beer, but only alcohol-free beer consumption was associated to a specific microbiota diversity with healthier function, suggesting that alcohol inhibited the positive effects of beer. As β-diversity was observed only after 30 days of treatment, the authors hypothesized that the effect on gut microbiota could depend on polyphenols and phenolic acids [111]. Similar results were obtained in an observational study, especially for higher butyric acid concentration in consumers versus non-consumers of beer [122], but no estimation of phenols intake was performed, nor were consumption of alcohol-free beer nor spirits-only drinkers recorded. On the other hand, another observational study on the microbiota of 916 UK female twins found association only for wine drinkers but not for beer (nor all other alcohols) [123], but also in this case, the consumption of alcohol-free beer was not considered. Figure 2 summarizes gut microbiota changes after beer consumption.

### 6.5. Role of Alcohol on Other Phenols-Related Effects of Beer 

In an observational follow-up prospective study (34 years) of the association between alcoholic beverage consumption (using repeated surveys) and dementia (*n* = 1462 women, 38–60 years), beer consumption was associated to reduced dementia risk compared to subjects consuming only spirits [125]. Unfortunately, the consumption of alcohol-free beer was not taken into account. Moreover, in a case-control study of the association, in postmenopausal women (*n* = 35,816; 55–69 years), between specific self-reported drinking/smoking habits (over 20 years) and diabetes, a reduction of risk was observed for moderate consumption of either beer, red or white wines, but also for liquor, making the authors disprove the hypothesis that flavonoids could protect from diabetes onset [127]. 

## 7. Fruit-Based Enrichment of Beer Phenols 

Beer is considered a promising beverage in the context of functional foods, which are food items with, in theory, health benefits, due to the enrichment with specific ingredients or bioactive compounds. Beer has high market opportunities because of an already high acceptancy of new organoleptic characteristics, due to widespread and previous diffusion of craft beers. Several ingredients have been added such as wheat, corn, rice and fruits. The phenolic profiles of several commercialized beers enriched with ingredients have already been reported and reviewed [8,133]. Studies agree that fruits’ refermentation and maturation within beer production is associated to a significant increase of flavors and bioactive compounds supporting benefits of fruit contribution to beer’s consumer acceptance. Both qualitative and quantitative increases in phenols have been reported in beers enriched with whole fruits during fermentation and works mainly focused on the role of the technological processes applied. However, rarely did a study report more than one fruit supplement. An exception is a recent report that compared individual phenols amounts in commercial beers enriched with cherry, raspberry, peach, apricot, grape, plum, orange or apple, and respective contribution to the antioxidant activity [134]. Importantly, this work demonstrates that fruit beers may be enriched with bioactive compounds (catechin, rutin, myricetin, quercetin and resveratrol) that are undetectable in conventional beers at identical extraction conditions and indicates enrichment with peels to be very promising because of the highest amount polyphenols and flavonoids content and antioxidant activity. Notably, resveratrol was found in beers enriched with all fruits except one (plum), with the highest level being measured in grape beer [134].

Other recent beer-added ingredients are quince fruit, mango, sweet potato and olive leaves. Because of organoleptic characteristics, quince fruit is specifically appreciated as a processed food. Many studies have shown that quince fruit lends itself as an affordable and good source of phenolic acids and flavonoids; in particular, in vitro assays have shown that phenols are the main compounds responsible for fruit’s hydrophilic antioxidant activity [135]. Quince fruit phenols have been extensively studied [136] and recent data indicate that the addition of different quince cultivars, with different sensory attributes or antioxidant content, can selectively modulate the final content in specific phenols and related sensory descriptors attributes [137]. The addition of quince increased the total polyphenol content, the total hydroxicinnamic acids, concentration of main volatile compounds related with fruity sensory descriptors, and led to higher intensities of floral and fruity sensory attributes [137]. The addition of mango fruit, naturally reach in phenols [138], yielded beers with higher polyphenol content and aroma than traditional beer, especially if the fruit was homogenized before addition, on the condition that no thermal treatment was performed [139].

The addition of dried flakes of sweet potato, naturally rich in phenols [140], before beer brewing increased both total phenols (about 10%) and flavonoids (about 20%) content without changing physicochemical and sensory parameters of beer, which also benefited from an important increase in *β*-carotene [141].

Also, dried olives or resulting extracts, that contain not only common phenols, but also the olive tree family-exclusive secoiridoids [142], were added to beer, and the resulting beer had positive flavor and aroma, but low colloidal stability and showed increasing haze formation during storage due to very high polyphenols content [143]. Similar increase in colloidal haze was reported also for beers with added omija fruits, questioning the validity of increasing the phenolic content of beers too much [144]. Beers enriched with lignans from wood chips or extracts displayed excessive bitter taste and unusual resin aroma, indicating the need for technological approaches to avoid significant changes to the characteristics of beer. A possible solution could come from the use of hot water as a unique solvent, already applied for the removal of resins from wood chips or lignan extracts from the knots of spruce trees (*Picea abies*), a strategy that yielded beer with as much as 100 mg/L of lignans.

## 8. Cereal-Based Enrichment of Beer Phenols 

Apart from barley, other malted cereals have been used since antiquity for the development of fermented beverages, in a somehow geographical way, for example, rice in India [145], millet in Nigeria [146], sorghum in South Africa [147] and Corn in Mexico [148]. Regarding the latter, a pulque-fermented drink known as “Sendechó” was antiquely prepared by the Mazahuas population in the Valley of Mexico, using chili and pigmented corn varieties with high content of phenolic compounds, mainly anthocyanins [149], which are completely absent in barley beer. In the attempt of developing a beer with traditional ingredients, pulque was substituted with hop and brewer’s yeast in an ale fermentation process performed with guajillo chili and blue corn malt. The result was a beer with total polyphenols concentration up to 560 mg GAE/L and of total anthocyanins up to 19.4 mg cyaniding-3-glucoside/L [150]. More recently, the same laboratory obtained blue or red corn malt blended beers with even higher total phenols amount (up to 849.5 mg GAE/L) and identified anthocyanins responsible for the final color yield of red and blue corn beers (pelargonidin-3-glucoside and cyanidin-3-glucoside) [151]. Authors also identified, in corn beers both previously reported and unreported, volatile phenols conferring desirable aromas to beers. Such results are promising with respect to previous reports of lower content of phenols in corn-added beers [152]. Nevertheless, to our knowledge, no consumer acceptability of such beers has been evaluated, and this aspect is crucial especially for the high content of phenols that can contribute to high spicy perception and for astringency of anthocyanins [151]. Among other gluten-free beers, those obtained from oat [153], sorghum [147], teff [154], millet [146], buckwheat [155] and quinoa [156] are in theory valid alternatives in terms of phenols considering the grain natural content, even if very little is available on the phenolic content of such beers, i.e., only a sum of aromatic alcohols was reported for millet [157]. A noteworthy emerging exception is represented by indigenous beer-like fermented beverages “ikigage” [158], “burukutu” and “pito” [159], drinks for which 4-vinylphenols quantities have been reported. Another exception is that of rice-based alcoholic beverages of Assam, India, with total polyphenol content up to 631.33 mg GAE/L [160].

One frequent issue of non-barley cereals is the low diastatic power, that traps phenols, making necessary the combination with other cereals or the addition of exogenous enzymes [161]. Addition to the mashing process of recombinant ferulic acid esterase [162] was recently proven as a valid remedy also for the low amount of the desirable phenol 4-vinylguaiacol (derived from ferulic acid by enzymic decarboxylation), a common issue of top-fermented wheat beer [163]. More recently, the strategy was further implemented by producing yeasts expressing bacterial ferulic acid decarboxylase [164].

## 9. Phenols in Non-Alcoholic and Isotonic Beers

Driving laws and a healthier lifestyle have increased the popularity of non-alcoholic beers. In order to not exceed the limit of 0.5% (*v*/*v*) alcohol or to produce beer with a limited alcohol content, two approaches are exploited. The first one consists in limiting the fermentation process, and hence the alcohol production, using low-alcohol yeasts or producing a wort with low degrees Plato and low diastatic power in order to obtain more dextrins than fermentable sugars. The alternative approach involves physical methods to remove the alcohol at the end of brewing, for example by vacuum evaporation or reverse osmosis treatments. Unfortunately, limiting the fermentation process can bring about inadequate conversion of wort to beer and, on the other hand, physical methods for alcohol removal can deteriorate beer composition [165]. Osmotic distillation using a membrane contactor was recently shown to be able to maintain the total phenols content in a low-alcohol top-fermented beer [166]. On the other hand, using the fermentation interruption approach, De Fusco and coworkers recently obtained a low-alcohol isotonic beer with an amount of total phenolic compounds similar to that of Pilsen beer and sport drinks [167]. Isotonic beers are an improvement on low-alcohol beers with similar rehydration potential of sports drinks [168] (beverages with specific osmolality and carbohydrate content [169]) and with the advantage of containing bioactive molecules. Notably, De Fusco and coworkers found that fermentation interruption did not significantly affect total phenols level [167]. Nevertheless, experiments are needed to test the shelf-life of low-alcohol isotonic beers and specifically to test if phenols’ antimicrobial activity is adequate in such low-alcoholic and carbohydrate-containing beverages [170].

## 10. Future Directions and Conclusions

Here, we attempted to review the more recent findings on beer phenols and their role in human health. Particular attention was dedicated to the role of genetic factors and to the enrichment with phenolic compounds by cereals different from barley or fruits naturally rich in phenols. In this respect, it would be interesting to investigate to what extent fruit addition also increases the alcoholic content, which has health and consumer acceptance consequences that are not negligible. One other interesting question regards the huge amount of debris produced by beer production, especially in terms of phenolic compounds (1% in by-product spent grain [171]) that could be recycled for beer enrichment itself. Recent reports indicate that the recovery of phenols can be improved using the fungi *Rhizopus oligosporus* as a fermenting organism [172].

Even if the Scopus.com search string we used is arbitrary and may not entirely represent the research on health effects of beer ascribable to the presence of phenolic compounds, less than of 25% (40 out of 161) of entries we retrieved were reports on the in vivo (human or animal) or in vitro effects of phenolic compounds within in toto beer. Most of the research is focused on evaluating the effects of single phenolic compounds of beer, which, however, can give rise to partial conclusions that need further experiments performed in physiological conditions. For example, as beer contains a mean amount of xanthohumol around 0.2 mg/L, what is the rationale for supplementing volunteers with an enriched drink containing a daily dose of 12 mg of xanthohumol [173], corresponding to the amount found in 60 L of beer? Indeed, several techniques have been used, starting from almost 20 years ago and featuring a patented addition of an enriched hop product [174], in order to increase the amount of this compound in beer up to 10 mg/L [175,176,177]. However, even if studies with enriched beers are helpful for assessing the metabolic fate of phenolic compounds, in order to correctly evaluate healthy effects of beer consumption, researchers should consider, besides the side effects of alcohol (where ethically possible), also those possibly due to yet uncharacterized molecules, i.e., those resulting from the addition of the enriched hop product, and those due to a non-physiological consumption of a single flavonoid for which pro-apoptotic effects are already known [178]. In fact, the only health claims authorized for phenolics by the European Food Safety Authority regard, at the moment, olive oil hydroxytyrosol and cocoa flavanols, with high daily amounts (5 and 200 mg, respectively) that can, however, be easily consumed in the context of a balanced diet [179,180]. Among retrieved reports, only six investigated the effects of phenols in the presence and absence of alcohol, thus also considering the effects of alcohol alone. Actually, four publications belong to the same Spanish research group [112,113,114,115], and thus probably refer to the same and unique small population of 33 high-cardiovascular-risk males.

In conclusion, studies applying a parallel administration of non-alcoholic beer or/and alcohol alone, in both animal and human intervention studies, support the existence of somehow interfering effects of phenols and ethanol. However, in order to better highlight additive or synergistic effects, further correctly set-up human interventional crossover or observational, or at least animal, studies are required. From this point of view, even insect models could deserve more attention. In fact, using *D. melanogaster* fed with a total beer extract, Merinas-Amo and colleagues were able to demonstrate the synergic interaction between different molecules contained in beer [181].

## Figures and Tables

**Figure 1 molecules-26-00486-f001:**
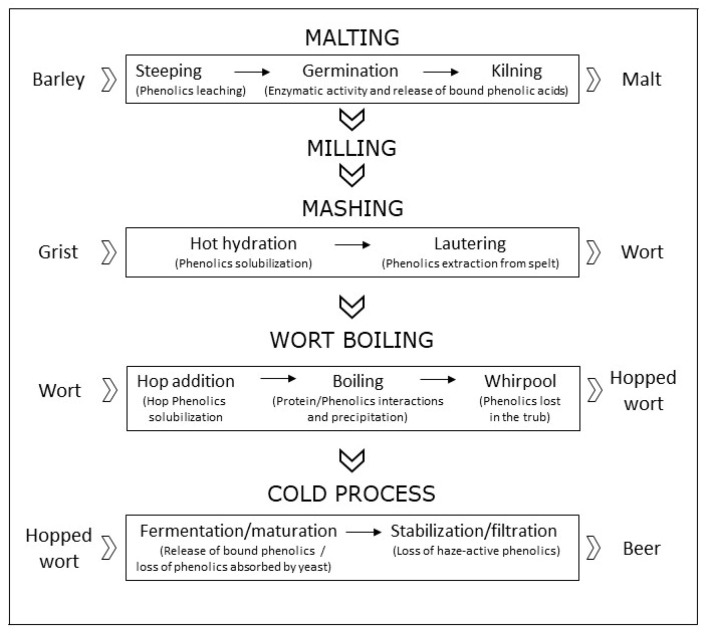
Phenolic compounds’ fate during the phases of malting and brewing processes: in the phase of mashing, after an initial decrease, total phenolics amount increases 3- to 5-fold; afterwards, phenolics continue to increase throughout mashing and during hop addition, but dramatically decrease during wort boiling, whirpool, fermentation, maturation, stabilization and filtration, so that, during the entire brewing process, about 60% of the malt phenolic content is lost.

**Figure 2 molecules-26-00486-f002:**
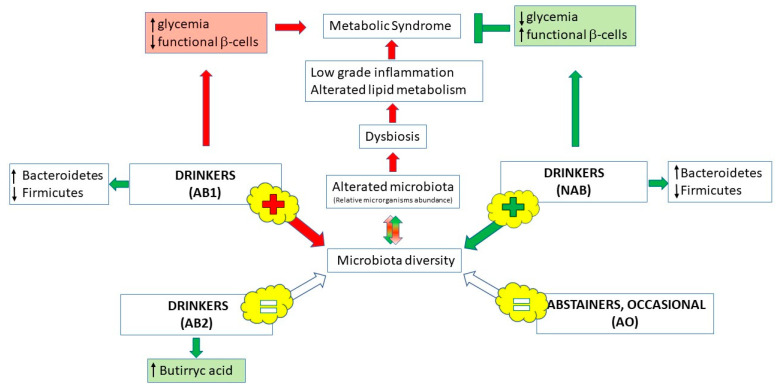
Schematic representation of relationship between beer, gut microbiota and metabolic syndrome. Phenolic compounds contained in non-alcoholic beer have a positive effect on the microbiota dysbiosis, one of the main causes of metabolic syndrome, but the effect is prevented by alcohol presence. Non-alcoholic beer consumption also determines a positive modification of some parameters typical of metabolic syndrome such as glycemia and the β-cells’ function (AB1, drinkers of 355 mL/day of alcoholic beer; NAB, drinkers of 355 mL/day of non-alcoholic beer [111]). On the other hand, moderate beer consumption can increase the production of butyric acid, a fundamental molecule produced by the microbiota and useful for its healthy implications (AB2, drinkers of 200–600 mL/day; AO, abstainers or occasional consumers of <1.5 alcohol g/day [122]).

**Table 1 molecules-26-00486-t001:** In vitro and animal studies.

Experimental Model	Tested Parameters	Observations	Non-Alcoholic Beer	Alcoholic Beer	Ethanol	References
in vitro, rat C6 glioma and human SH-SY5Y neuroblastoma cells, treated with total extracts from dark, non-alcoholic or lager beers	cell viability and adenosine receptors gene expression and protein levels following oxidant stressor (hydrogen peroxide) challenge	alcoholic dark beer extract conferred higher protection compared to lager or non-alcoholic beer extracts	yes	no	no	[91]
animal, 36 prepuberal Wistar rats fed with beer or ethanol (both 10%) or water for 2/4 weeks	plasma reproductive hormones, cleaved caspase-3 immunolocalization and neuronal nitric oxide synthase level in Leydig cells	beer decreased sex hormones compared to ethanol or water rats and inhibited ethanol-induced increase of cleaved caspase-3	no	yes	yes	[92]
animal, 70 male Wistar rats, with monocrotaline-induced pulmonary arterial hypertension, fed with xanthohumol-fortified beer or ethanol (both 5.2%) for 4 weeks	cardiopulmonary exercise testing and hemodynamic recordings, analysis of pulmonary vascular remodeling and cardiac function	xanthohumol-fortified beer attenuated pharmacologically induced pulmonary vascular remodeling and improved cardiac function, compared to ethanol rats	no	yes	yes	[93]
animal, 40 male Wistar rats, with aluminium nitrate-induced inflammatory status, fed with low alcoholic-beer (0.9%) or hops or silicons for 3 months	animal behavior and brain antioxidant and anti-inflammatory status	non-alcoholic beer, but also silicon and hops alone, prevented aluminum-induced inflammation and neurodegenerative effects	yes	no	no	[94]
animal, 30 male Wistar rats, with streptozotocin-induced diabetes, fed with alcoholic beer or xanthohumol-enriched or 5% ethanol for 5 weeks	hepatic glucolipid metabolism, levels lipogenic enzymes and glucose transporter 2	alcoholic beer enriched with xanthohumol (but not normal beer nor ethanol) prevented the streptozotocin-induced liver catabolic state alterations	no	yes	yes	[95]
animal, 30 male Wistar rats, with skin induced wound healing and streptozotocin-induced diabetes, fed with alcoholic beer or xanthohumol-enriched or 5% ethanol for 5 weeks	effects on wound healing, through evaluation of angiogenesis, inflammation and oxidative stress modulation	alcoholic and xanthohumol-enriched beers respectively, prevented and reversed the alcohol-induced markers of inflammation, oxidative stress and angiogenesis	no	yes	yes	[96]
animal, 24 male Wistar rats, with skin induced wound healing, fed with xanthohumol-fortified alcoholic beer or 5% ethanol for 4 weeks	angiogenesis and inflammation markers (serum vascular endothelial growth factor levels, *N*-acetylglucosaminidase activity, Interleukin-1 β concentration)	alcoholic and xanthohumol-enriched beer respectively, prevented and reversed the alcohol-induced markers of inflammation, oxidative stress and angiogenesis	no	yes	yes	[97]
in vitro, MKN-28 gastric epithelial cells, treated with different alcoholic beverages, at a similar ethanol concentration	tetrazolium (MTT) assay at 30, 60 and 120 min	alcoholic beer reduced cell viability like ethanol, while red wine, even dealcoholated, protected	no	yes	yes	[89]
animal, 32 spontaneously hypertensive and 32 normotensive Wister rats, fed intragastrically with lyophilized beer for 10 days	aminooxyacetic acid-induced γ-aminobutyric acid (GABA) accumulation in hypothalamus and pons-medulla	lyophilized beer decreased GABA accumulation	yes	no	no	[98]
animal, 36 male Wistar rats fed (4 weeks) with lyophilized, polyphenol-free, beer or white wine	plasma lipids and lipid peroxides	polyphenol-free beer (not polyphenol-free wine) significantly decreased lipids and lipid peroxides	no	yes	no	[99]
animal, 60 Wistar rats fed (4 weeks) with alcoholic (4%) or lyophilized beer	plasma lipids and lipid peroxides	both alcoholic and lyophilized beers similarly decreased lipids and lipid peroxides	yes	yes	no	[100]

**Table 2 molecules-26-00486-t002:** Intervention studies (*n*, subjects’ number; y, age (years)).

Experimental Model	Tested Parameters	Observations	Non-Alcoholic Beer	Alcoholic Beer	Ethanol	References
intervention trial (healthy adults), 30 days, 355 mL beer/day with (4.9%, *n* = 33, 21–55 y) or without alcohol (0.5%, *n* = 35, 21–53 y)	microbiota composition, fasting blood serum glucose, β-cell function	both beer interventions increased microbiota diversity, but only non-alcoholic beer increased heathier diversity and β-cells function and decreased fasting blood serum glucose	yes	yes	no	[111]
controlled clinical trial (healthy adults, *n* = 20, 18–45 y, single blind, randomized, crossover), single dose of beer (250 mL), with (4.5 or 8,5%) or without (0%) alcohol	urinary tyrosol (TYR) and hydroxytyrosol (HT)	non-alcoholic beer intervention increased HT recovery (and reduced TYR recovery) compared to alcoholic beer	yes	yes	no	[12]
intervention controlled trial (high cardiovascular risk males, *n* = 33, 55–75 y, open, randomized, crossover), 4 weeks, daily: 660 mL beer (1029 mg polyphenols and 30 g ethanol) or 990 mL non-alcoholic beer (1243 mg polyphenols and <1 g ethanol) or 100 mL gin (30 g ethanol)	urinary metabolomics	both beer intervention increased to similar extent urine excretion of hop α-acids and fermentation products, compared to gin intervention	yes	yes	yes	[112]
intervention controlled trial (high cardiovascular risk males, *n* = 33, 55–75 y, open, randomized, crossover), 4 weeks, daily: 660 mL beer (1029 mg polyphenols and 30 g ethanol) or 990 mL non-alcoholic beer (1243 mg polyphenols and <1 g ethanol) or 100 mL gin (30 g ethanol)	atherosclerotic and inflammation plasma biomarkers and peripheral blood mononuclear cells immunophenotyping	only non-alcoholic beer intervention reduced leukocyte adhesion molecules and inflammatory biomarkers, but alcoholic beer and gin interventions improved plasma lipid and atherosclerosis inflammatory markers	yes	yes	yes	[113]
intervention controlled trial (high cardiovascular risk males, *n* = 33, 55–75 y, open, randomized, crossover), 4 weeks, daily: 660 mL beer (1029 mg polyphenols and 30 g ethanol) or 990 mL non-alcoholic beer (1243 mg polyphenols and <1 g ethanol) or 100 mL gin (30 g ethanol)	number of circulating endothelial progenitor cells (EPC)	8-fold and 5-fold increases of EPC number respectively in alcoholic and non-alcoholic beer interventions and statistically not significant 5-fold decrease in gin administration	yes	yes	yes	[114]
intervention controlled trial (high cardiovascular risk males, *n* = 33, 55–75 y, open, randomized, crossover), 4 weeks, daily: 660 mL beer (1029 mg polyphenols and 30 g ethanol) or 990 mL non-alcoholic beer (1243 mg polyphenols and <1 g ethanol) or 100 mL gin (30 g ethanol)	urinary isoxanthohumol	beer administrations (not gin) induced similar excretion of urinary isoxanthohumol	yes	yes	yes	[115]
intervention trial (stressed healthy females, *n* = 17, 40.9 ± 10.5 y, randomized, crossover), 2 weeks 330 mL beer/day, first week non-alcoholic, second week alcoholic	antioxidant capacity in urine	non-alcoholic beer administration induced higher antioxidant capacity compared to alcoholic beer one	yes	yes	no	[110]
intervention trial (healthy males *n* = 17, 28.5 ± 5.2 y, randomized, single-blind, crossover), single dose (800 mL) beer (48 mg polyphenols and 20 g ethanol) or non-alcoholic beer (48 mg polyphenols) or vodka (20 g ethanol)	endothelial function, aortic stiffness, pressure wave reflections and aortic pressure	non-alcoholic and alcoholic beer interventions improved (similarly) arterial biomarkers but the effects were observed also for the vodka intervention alcoholic beer intervention improved wave reflections reduction better than vodka intervention	yes	yes	yes	[87]
intervention trial (postpartum breastfeeding-mother-infants dyads), 30 days 660 mL/day non-alcoholic beer (*n* = 30, 30 ± 5 y) or not (*n* = 30, 31 ± 3 y)	breastmilk, plasma and urine oxidative status	non-alcoholic beer increased breastmilk and plasma antioxidant capacities	yes	no	no	[107]
intervention trial (healthy male marathon runners, double-blind), 5 weeks (from 3 before to 2 after marathon) 1.0–1.5 L non-alcoholic beer (*n* = 142, 36–51 y) or control beverage without polyphenols (*n* = 135, 35–49)	blood inflammatory markers and upper respiratory tract illness (URTI) rates	non-alcoholic beer intervention reduced after-run blood inflammatory markers and URTI rates, compared to the polyphenols-free beverage	yes	no	no	[116]
intervention trial (healthy males, *n* = 10, 21–29 y, randomized, single-blind, crossover), single dose (7 mL/kg body wt) alcoholic beer (0.4 g/L GAE polyphenols and 0.32 g ethanol/kg body wt) or vodka (0.32 g ethanol/kg body wt)	plasma lipid peroxides, uric acid concentration and arterial stiffness following 100% O_2_ breathing-oxidative stress	alcoholic beer intervention protected against oxygen-induced increase in arterial stiffness but so did vodka	no	yes	yes	[106]
intervention (post-menopausal healthy females, *n* = 29, 64.5 ± 5.3 y, longitudinal), 45 days 500 mL alcoholic-free beer/day	lipid profile and plasma inflammatory markers	alcoholic-free beer intervention improved lipid profile and plasma inflammatory markers	yes	no	no	[108]
controlled clinical trial (hypercholesterolemic non-drinker males, *n* = 42, 43–71 y, randomized, single-blind), 30 days, daily: 330 mL 5.4% beer (20 g alcohol and 510 mg polyphenols) or water (containing beer mineral)	coronary atherosclerosis plasma markers	alcoholic beer intervention improved coronary atherosclerosis plasma markers compared to control administration water	no	yes	no	[117]
intervention (healthy adults, *n* = 10, 25–45 y, randomized), single dose (500 mL) 4.5% alcoholic beer	phenolic acids plasma metabolites	alcoholic beer intervention demonstrates absorption and metabolism of phenolic acids to glucuronide and sulfate conjugates	no	yes	no	[103]
intervention (healthy normotensive drinking men, *n* = 28, 20–65 y, randomized, crossover), 4 weeks, daily: 1125 mL 4.6% beer (41 g alcohol) or 375 mL 13% red wine 2023 mg/L polyphenols) or 375 mL dealcoholized red wine (2094 mg/L polyphenols)	blood pressure and vascular function following brachial artery flow-mediated and glyceryl trinitrate-mediated dilatation	alcoholic beer (but also wine) increased awake systolic blood pressure and asleep heart rate	no	yes	no	[118]
intervention (healthy adults, 25–45 y, randomized no crossover), single dose (500 mL): 4.5% alcoholic (*n* = 14) or dealcoholized beer or 4.5% ethanol (*n* = 7)	total plasma antioxidant status	alcoholic beer administration improved higher plasma antioxidant capacity compared to the dealcoholized one, thanks to higher absorption of phenolic acids	yes	yes	yes	[104]
intervention (healthy males, *n* = 5, 23–40 y), single dose (4 L) low-alcohol (1%) beer	urinary ferulic and its glucuronide	beer administration demonstrates bioavailability of ferulic acid	yes	no	no	[102]
intervention trial (healthy male drinkers, *n* = 27, 49.2 ± 2.3 y, randomized, crossover), 4 weeks, daily 375 mL: 4.9% or 0.9% beer (similar phenolic content 310–330 mg/L)	LDL in vitro oxidizability and characterization	switch from low to high alcoholic beer intervention increased LDL oxidizability	yes	yes	no	[109]

**Table 3 molecules-26-00486-t003:** Observational studies (*n*, subjects’ number; y, age (years)).

Experimental Model	Tested Parameters	Observations	Non-Alcoholic Beer	Alcoholic Beer	Ethanol	References
observational (ALMICROBHOL adults *n* = 78, 25–50 y), alcoholics BCQ	microbiota composition (16S rRNA sequencing) and short chain fatty acid profile in fecal samples	higher butyric acid concentration and gut microbial diversity in consumers vs. non-consumers of beer	no	yes	no	[122]
observational (TwinsUK females *n* = 916, 16–98 y), alcoholics FFQ	microbiota composition in fecal samples (16S rRNA sequencing)	no association between beer (nor all alcohols except wine) consumption and gut microbial diversity	no	yes	yes	[123]
observational (MEAL Southern Italy adults, *n* = 2044, >18 y), phenolics FFQ	hypertension (arterial blood pressure measurement)	inverse association between beer consumption and hypertension	no	yes	no	[121]
observational prospective (2002–2003 CMHS Californian males *n* = 84,170, 45–69 y), alcoholics FFQ	prostate cancer registries (Surveillance Epidemiology and End Result)	no association between beer (nor wine nor liquor) consumption and prostate cancer	no	yes	yes	[124]
observational cross-sectional (IMMIDIET Italy–Belgium–UK female–male pairs, *n* = 1604, 26–65 y), alcoholics FFQ	plasma and red blood cell omega–3 fatty acids	no association between beer consumption and plasma or red blood cell omega 3 fatty acids (reduced for wine)	no	yes	no	[119]
observational 34 year prospective (PPSWG Sweden females, *n* = 1462; 38, 46, 50, 54, 60 y), alcoholics BCQ	dementia (neuropsychiatric years-repeated examinations)	direct association between beer (or wine) consumption and longevity and reduced dementia risk (compared to subjects consuming only spirits)	no	yes	yes	[125]
observational (over 10 years) case-control matched leukoplakia subjects (*n* = 187 + 187, 40–65 y), alcoholics FFQ	leukoplakia (clinical examination and biopsy)	no significant association between moderate beer drinking and leukoplakia risk (increased for spirit and reduced for wine)	no	yes	yes	[126]
observational case-control prospective (1987–2004 IWHS) diabetes postmenopausal females (*n* = 35,816; 55–69 y), flavonoids FFQ	self-reported diabetes	inverse association between beer (or other alcoholic beverages including liquor) consumption and diabetes risk	no	yes	yes	[127]
observational oral cancer mortality rate (20 Nations male 2002 age-standardized), national mean alcoholic beverage consumption	oral cancer mortality rates (International Agency for Research on Cancer)	no association between beer (nor wine, but association for spirits) consumption alone and oral cancer risk	no	yes	yes	[128]
observational case-control matched (1993–1996 King County, WA) prostate cancer subjects (*n* = 753 + 703; 40–64 y), alcoholics BCQ	prostate cancer registry (Seattle Puget Sound Surveillance Epidemiology and End Results Cancer Registry), histological confirmation	no association for beer consumption (nor liquor but association for wine) and prostate cancer risk	no	yes	yes	[129]
observational case-control matched prospective (1980–1993 Québec) child acute lymphoblastic leukemia (*n* = 491 + 491; 0–9 y), parents alcoholics BCQ	child acute lymphoblastic leukemia hospital diagnosis	inverse association between mothers’ beer (but not spirits) consumption and child acute lymphoblastic leukemia	no	yes	yes	[130]

FFQ, food frequency questionnaires; BCQ, beverage consumption questionnaires; ALMICROBHOL, Effects of Alcohol Consumption on Gut Microbiota Composition in Adults; TwinsUK, UK Adult Twin Registry; MEAL, Mediterranean healthy Eating, Ageing, and Lifestyle; CMHS, California Men’s Health Study; IMMIDIET, Dietary Habit Profile in European Communities with Different Risk of Myocardial Infarction: the Impact of Migration as a Model of Gene-Environment Interaction; PPSWG, The Prospective Study of Women in Gothenburg; IWHS, Iowa Women’s Health Study.

## Data Availability

Not applicable.

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
