# Peer review of "The Role of Bioactive Phenolic Compounds on the Impact of Beer on Health"

_molecules, 2021, doi:10.3390/molecules26020486_

Round 1
Reviewer 1 Report
In bold I have reported my suggestions to complete the sentences
In italics I have reported some considerations or questions
Line 72: the bitter taste. Hops affect only the bitter taste. Alternatively, it is more correct to indicate flavor or aroma
Line 75: yielding precipitated complexes responsible for the formation of chill haze
Line 86: What about pasteurization effect on the phenolic content of beer?
Line 112: in beer.
Line 159: just before the end of wort boiling
Line 160: Astringency, bitterness and fullness are also affected by the boiling time
Line 524: addition of exogenous enzymes
Line 527: wheat beer
Line 532: and hence the alcohol production, using low alcohol yeasts or producing a wort with low degrees Plato and low diastatic power in order to obtain more dextrins than fermentable sugars
Line 533: end of brewing, for example by vacuum evaporation or reverse osmosis treatments.
Line 564: If possible, provide more information regarding the daily dose of polyphenols to be taken in order to obtain a beneficial effect on health.
Author Response
Dear Reviewer, please find below the point-by-point answers (green colour) to your comments (please see attached doc file):
Line 72: the bitter taste. Hops affect only the bitter taste. Alternatively, it is more correct to indicate flavor or aroma
The word "bitter" was added (line 124)
Line 75: yielding precipitated complexes responsible for the formation of chill haze
Corrected as suggested (line 127) and further modified following the insertion of Figure 1
Line 86: What about pasteurization effect on the phenolic content of beer?
A sentence and a reference to a review has been added (line 139). We thank the reviewer for noticing
Line 112: in beer.
Corrected (line 70)
Line 159: just before the end of wort boiling
Corrected (line 176)
Line 160: Astringency, bitterness and fullness are also affected by the boiling time
We fully agree with the Reviewer and corrected the text mentioning the same ref of the comment on pasteurization (line 177)
Line 524: addition of exogenous enzymes
Corrected (line 556)
Line 527: wheat beer
Corrected (line 559)
Line 532: and hence the alcohol production, using low alcohol yeasts or producing a wort with low degrees Plato and low diastatic power in order to obtain more dextrins than fermentable sugars
Corrected (line 565). We thank the reviewer for the observation
Line 533: end of brewing, for example by vacuum evaporation or reverse osmosis treatments.
Corrected (line 568)
Line 564: If possible, provide more information regarding the daily dose of polyphenols to be taken in order to obtain a beneficial effect on health.
Done (line 608)
Reviewer 2 Report
The authors present a systematic (health-related effects) and narrative (other sections) review on beer-derived polyphenols and the effect of beer production parameters on phenolic stability/occurrence and subsequent health benefits. This review has a quite interesting historical background and the systematic searching was concrete, although relevant review articles on this theme were not included (e.g. doi: 10.1080/87559120903157954, 10.3390/molecules24081568 , 10.3390/molecules25214960 , 10.1016/j.fct.2020.111782). Some modifications are suggested to improve the manuscript´s scientific soundness and uniqueness.
General comments:
- The manuscript will improve substantially if it is reviewed by someone whose native language is English. Syntaxis is particularly and issue (e.g. lines 24-26 could be “Annual per capita consumption (L/year) of beer has sharply increased in Czech Republic (141 L), US (50-80) and France (33)”
- One or two images are strongly suggested
Title. OK.
Abstract. OK, although it should reflect the differential scientific contribution of this new review article and the way it is organized.
Introduction. OK
Body. A graph depicting all critical steps during malting and brewing that lead to polyphenol loss (Section 2) and the most promising healthy polyphenols (Section 3) will be helpful.
- Rearrangement of Section 3 before sections 2+4 (merge into one section) is suggested. Section 2.
- A more in-depth discussion on the effect of losing polyphenols during malting/brewing on aroma and flavor (doi:10.3390/fermentation4010020) since some types of brewing processes (and so their phenolic content) has gained recognition in the last decade [e.g. Czech Beer; Olšovská et al. (2014) Czech J. Food Sci. 2014, 32(4): 309).
- A brief comment on the effect of adding other phenolic substrates in later brewing steps on the polyphenol content and antioxidant activity of artisanal beer, could be expanded (use reference 10, line 49 and similar) and transfer sections 7 and 8 afterward.
- Expanding the “symbiotic” effect of beer in section 6.4 by adding a Table or Figure of gut microbiota changes after beer consumption (or brewers vs. non-brewers), is strongly suggested.
- Tables. Transfer each table to the corresponding sections and modify lines 252-256 as follows: “For the remaining 41 (minus one not available even by the authors themselves [88]), experimental models, parameters tested, and main findings are summarized and sorted chronologically in the following sections”.
Minor comments:
- Do not forget to describe the meaning of each abbreviation the first time it is mentioned for those who were left in the text.
- Reduce as much as possible references older than 10y to say 25%
Author Response
Dear Reviewer, please find below the point-by-point answers (green colour) to your comments (please see attached doc file):
General comments:
- The manuscript will improve substantially if it is reviewed by someone whose native language is English. Syntaxis is particularly and issue (e.g. lines 24-26 could be “Annual per capita consumption (L/year) of beer has sharply increased in Czech Republic (141 L), US (50-80) and France (33)”
The sentence has been replaced as suggested (line 27) and English has been checked thoroughly
- One or two images are strongly suggested
One image depicting phenolics loss and one more depicting "symbiotic" effects have been added
Title. OK.
Abstract. OK, although it should reflect the differential scientific contribution of this new review article and the way it is organized.
The section on health-related effects is now better mentioned in the abstract (line 15)
Introduction. OK
Body. A graph depicting all critical steps during malting and brewing that lead to polyphenol loss (Section 2) and the most promising healthy polyphenols (Section 3) will be helpful.
Figure 1, containing a gragh depicting loss of phenolics during processing, has been added and the text of the entire section 3 was modified accordingly (lines 85-139)
- Rearrangement of Section 3 before sections 2+4 (merge into one section) is suggested. Section 2.
Section 3 has been moved before section 2, however section 4 has been kept separated for easier reading
- A more in-depth discussion on the effect of losing polyphenols during malting/brewing on aroma and flavor (doi:10.3390/fermentation4010020) since some types of brewing processes (and so their phenolic content) has gained recognition in the last decade [e.g. Czech Beer; Olšovská et al. (2014) Czech J. Food Sci. 2014, 32(4): 309).
Loss of phenols during processing is now better discussed and a figure has been added (Figure 1). Moreover Czech beers are now mentioned (lines 161-163). We thank the reviewer for pointing out this lack in the first submission
- A brief comment on the effect of adding other phenolic substrates in later brewing steps on the polyphenol content and antioxidant activity of artisanal beer, could be expanded (use reference 10, line 49 and similar) and transfer sections 7 and 8 afterward.
Hoping to meet Reviewer's suggestion, a brief comment on enrichment of beer with xanthohumol using hop products was included in the discussion, and the text updated accordingly (lines 599-609)
- Expanding the “symbiotic” effect of beer in section 6.4 by adding a Table or Figure of gut microbiota changes after beer consumption (or brewers vs. non-brewers), is strongly suggested.
Figure 2, describing microbiota changes, has been added
- Tables. Transfer each table to the corresponding sections and modify lines 252-256 as follows: “For the remaining 41 (minus one not available even by the authors themselves [88]), experimental models, parameters tested, and main findings are summarized and sorted chronologically in the following sections”.
Tables were moved and text changed as suggested (exception for the fact that the chronological sorting is present entirely only in tables) (lines 270-271)
Minor comments:
- Do not forget to describe the meaning of each abbreviation the first time it is mentioned for those who were left in the text.
Done
- Reduce as much as possible references older than 10y to say 25%
Done (5 references have been deleted)